# Use of Preliminary Exposure Reduction Practices or Laundering to Mitigate Polycyclic Aromatic Hydrocarbon Contamination on Firefighter Personal Protective Equipment Ensembles

**DOI:** 10.3390/ijerph20032108

**Published:** 2023-01-24

**Authors:** Andrea F. Wilkinson, Kenneth W. Fent, Alexander C. Mayer, I-Chen Chen, Richard M. Kesler, Steve Kerber, Denise L. Smith, Gavin P. Horn

**Affiliations:** 1Division of Field Studies and Engineering, National Institute for Occupational Safety and Health (NIOSH), Centers for Disease Control and Prevention (CDC), Cincinnati, OH 45226, USA; 2Fire Safety Research Institute, UL Research Institutes, Columbia, MD 21045, USA; 3Health and Human Physiological Sciences, Skidmore College, Saratoga Springs, NY 12866, USA; 4Illinois Fire Service Institute, University of Illinois at Urbana-Champaign, Urbana, IL 61801, USA

**Keywords:** decontamination, polycyclic aromatic hydrocarbons (PAHs), volatile organic compounds (VOCs)

## Abstract

Chronic health risks associated with firefighting continue to be documented and studied, however, the complexity of occupational exposures and the relationship between occupational exposure and contaminated personal protective equipment (PPE) remains unknown. Recent work has revealed that common PPE cleaning practices, which are becoming increasingly more common in the fire service, are not effective in removing certain contaminants, such as polycyclic aromatic hydrocarbons (PAHs), from PPE. To better understand the relationship between contaminated firefighter PPE and potential exposure to PAHs, and to gain further understanding of the efficacy of cleaning practices, we used a standardized fire exposure simulator that created repeatable conditions and measured PPE surface contamination levels via wipe sampling and filters attached to firefighter gear worn by standing mannequins. This study examined the effects of repeated (40 cycles) PPE cleaning (laundering and on-scene preliminary exposure reduction (PER) techniques) and repeated exposures on PAH concentration on different surfaces. Further exploration included examination of contamination breakthrough of turnout jackets (comparing outer shell and interior liner) and evaluation of off-gassing PAHs from used gear after different cleaning treatments. When compared by jacket closure type (zipper and hook and dee), total PAH concentration wiped from gear after exposure and cleanings showed no significant differences. Regression analysis indicated that there was no effect of repeated exposures on PAH contamination levels (all sampling sites combined; before fire 10, 20, and 40; after fire 1, 10, 20, and 40; *p*-value > 0.05). Both laundering and on-scene PER significantly reduced contamination levels on the exterior pants and helmets and were effective at reducing PAH contamination. The jacket outer shell had significantly higher PAH contamination than the jacket liner. Both laundering and wet soap PER methods (post-fire) are effective in reducing surface contamination and appear to prevent accumulation of contamination after repeated exposures. Semi-volatile PAHs deep within the fibers of bulky PPE are not effectively reduced via PER or machine laundering, therefore, permitting continued off-gassing of these compounds. Further research is needed to identify the most effective laundering methods for firefighter turnout gear that considers the broad spectrum of common contaminants.

## 1. Introduction

The acute risks of structural firefighting have been relatively well documented. However, many of the chronic health risks associated with complex occupational exposures, the relationship between exposure risk and contaminated personal protective equipment (PPE), and the effectiveness of contamination control practices remains unknown. PPE, such as the turnout gear ensemble traditionally worn by firefighters, becomes contaminated following emergency responses through exposure to products of combustion. The vast variety of chemical compounds firefighters can be exposed to during fire suppression activities or overhaul work has received attention from various researchers [1,2,3,4]; however, the best practices for cleaning and risk mitigation continue to evolve, specifically as they pertain to the handling of contaminated PPE and cross-contamination exposure risks. It is possible that the effectiveness of preliminary exposure reduction (PER) may vary by contaminants or classes of contaminants.

Polycyclic aromatic hydrocarbons (PAHs), which are products of incomplete combustion and include several semi-volatile organic compounds (SVOCs), have routinely been found to be elevated on the PPE of firefighters [1,5]. PAHs are a known occupational exposure hazard [6] and have gained recent attention in the fire service due to their association with an increased risk of cancer [7,8]. Multiple studies have shown that firefighters have an increased risk for developing certain types of cancers, including melanoma, testicular, bladder, prostate, colorectal, leukemia, and lung cancer [7,9].

The International Association for Research on Cancer (IARC) acknowledged the exposure risks of firefighting and classified the occupation of firefighting as Group 1, carcinogenic to humans, based on evidence available in 2022 [8,10,11]. PAH exposure is an important factor that was considered by the IARC monograph working group in 2010 and again in 2022 [11,12]. Not all PAHs have similar toxicity and therefore it is important to evaluate toxicity equivalencies (TEQ) using methods such as the model created by Nisbet and LaGoy [5]. This model is intended for related compounds to be ranked by toxicity and carcinogenic potential and is based on one of the more toxic PAH compounds, benzo[a]pyrene. At least one study examining firefighter exposures has incorporated this or similar TEQ assessments [13,14].

Firefighting PPE has been designed primarily to provide thermal protection, but also provides an important level of protection against potential hazards, such as abrasions, body fluids, and exposure to combustion byproducts. However, PAHs and other chemicals have been found to contaminate PPE and skin [15,16]. Cross-transfer of PAHs from PPE to skin is particularly relevant during PPE doffing [17]. PAHs that transfer to the skin could be absorbed or ingested [18]. Of additional concern is the off-gassing of compounds on contaminated gear that can potentially be inhaled after firefighting [19,20]. Although some previous work has examined the efficacy of PPE cleaning practices for PAH contamination, little has been published on the quantification of PAHs embedded deep in the PPE fabric that may slowly off-gas [16,21,22]. Recent work has revealed that common PPE cleaning practices are not effective at removing certain contaminants (including some types of SVOCs) from areas such as the moisture barriers in jackets and pants, hoods, and other components of PPE [6,23,24,25].

To mitigate these exposure risks, many fire departments have started engaging in new and more frequent PPE cleaning protocols. Gross on-scene decontamination, more appropriately referred to as preliminary exposure reduction (PER), is sometimes used as an initial cleaning step and is then followed by bagging and laundering of the turnout gear. There are many different types of decontamination processes that may be employed, which can be grouped into those that use water (e.g., wet soap cleaning) or those that can be performed while keeping the PPE dry (e.g., dry brush cleaning) [16]. Wet soap cleaning has been shown to be more effective (cite earlier FSI study) and is recommended as ‘preliminary exposure reduction’ by NFPA 1851: Standard on Selection, Care, and Maintenance of Protective Ensembles for Structural Fire Fighting and Proximity Fire Fighting [26]. Even departments that rely primarily on PER for PPE cleaning to remove contaminants from gear will routinely launder their turnout gear (e.g., once or twice per year per manufacturer recommendations and NFPA standard 1581) [26]. Numerous studies have evaluated the life cycles of PPE and laundering effects [22,27,28]; however, the emphasis of these studies was to evaluate longevity of PPE performance properties. To our knowledge, no studies have been conducted to evaluate routine laundering or PER of PPE as way to manage cross-contamination or potential for accumulation of contaminants over time. Fent et al. [16] found that PAH contamination on firefighter PPE increased with each use, but this was without any cleaning treatments.

For this study, we chose to focus on PAH contaminants because we were certain of their production during the experiments and confident in our ability to measure them on surfaces and in air. Other hazardous substances released during combustion such as per- and polyfluoroalkyl substances (PFAS) may contaminate turnout gear, but this depends on the fuel package and treatments (e.g., stain resistance coatings, which were not included in the experiments) and was beyond the scope of this study. To better understand the relationship between contaminated firefighter PPE and potential exposure to PAHs, and to gain further understanding of the efficacy of cleaning practices, we used a standardized fire exposure simulator that created repeatable conditions and measured PPE surface contamination levels via wipe sampling and filters attached to firefighter gear worn by standing mannequins. Specifically, this study examined the effects of PPE repeated (40 cycles) cleaning techniques (laundering and on-scene preliminary exposure reduction practices) and repeated exposures (40) in the fireground exposure simulator (FES) on PAH concentration on different surfaces. TEQ analysis was also utilized to evaluate toxicity equivalencies. Further exploration included examination of contamination breakthrough of turnout jackets (comparing outer shell and interior liner) and evaluation of off-gassing PAHs from used gear after different cleaning treatments.

## 2. Materials and Methods

This study was performed at the University of Illinois Fire Service Institute (IFSI) with collaboration from the National Institute for Occupational Safety and Health (NIOSH) and the Fire Safety Research Institute (FSRI).

### 2.1. Fireground Exposure Simulator

The FES and study design are described in detail elsewhere [17,29]. Briefly, mannequins were dressed in firefighting PPE and placed in the FES prop (Figure 1). The FES prop is a 2.4 m wide, 2.9 m tall, and 12.2 m long steel intermodal shipping container (85 m^3^), divided into three sections, where fire effluent from the 3.1 m long (center section) combustion chamber was ducted into the two 4.6 m long exposure chambers (east and west). The FES was designed to allow for reproducible fire conditions.

Up to 11 full-sized mannequins dressed in firefighting PPE ensembles were placed between the east and west exposure chamber. A popular residential style sofa was ignited in the combustion chamber and mannequins were exposed to the smoke from the burning sofa for ten minutes. This process of exposure (and subsequent PPE cleaning) was repeated for a total of 40 trials. The protocol timing for ignition, ventilation, and suppression was standardized following guidelines from previous fireground research [30]. The sofas for each burn were identical and purchased for the purpose of research, thereby ensuring a similar rate of combustion and product release.

### 2.2. Firefighting Personal Protective Equipment

The full-sized mannequins were dressed in commercially available NFPA 1971 third party certified compliant firefighting PPE ensembles (jacket, pants, boots, hood, helmet, and gloves) [31]. This study included two different turnout jacket closure options; one set of jackets utilized a zipper and Velcro^®^ closure while the second utilized a hook and dee closure. Having two different jacket closures allowed us to compare the effectiveness and amount of contaminant breakthrough of each closure. Both sets of turnout jackets were equipped with the same outer shell (Kevlar^®^/Nomex^®^), moisture barrier (ePTEE), and thermal layer (Kevlar^®^/Lenzing FR^®^ face cloth with Nomex^®^ batting). The materials and design of the PPE ensemble were selected due to popularity at the time of this study. The mannequins were further outfitted with a self-contained breathing apparatus (SCBA) facepiece, a belt to simulate the waist strap of SCBA, a hood, helmet, gloves, and firefighting boots.

### 2.3. PPE Cleaning Treatments

Following each ten-minute exposure trial, turnout jackets were either cleaned via on-scene wet soap PER, a method intended to reduce surface contamination prior to full cleaning, or machine laundering following NFPA 1851 guidelines [26]. PPE was assigned to one of three groups; (1) zippered PPE, laundered following the manufacturer’s recommendations (*n* = 3), (2) zippered PPE cleaned via on-scene wet soap decontamination (*n* = 3), or (3) hook and dee PPE, laundered following the manufacturer’s instructions (*n* = 3). Zippered jackets were laundered separately from hook and dee jackets (Appendix A).

Laundering of PPE followed the NFPA 1851-2014 guidelines (current edition at the time of the study), utilizing a front-loading extractor (MWR27X5 Gear Guardian, Milnor), warm water, and Patriot Chemical Firehouse Detergent (NFPA 2014). Following washing, PPE was hung on racks in a forced ambient air cabinet (ADFG-56, ADC Gear Drying Cabinet) to facilitate air circulation. Jacket liners and shells were separated and all components (liners, shells, and pants) turned inside out so the moisture barrier was on the inside and thermal layer on the outside. Shells were fastened via zipper or hook and dee and washed right-side out. Separate laundering machines were utilized for each PPE component throughout the study (liners were washed separately from shells).

PER occurred via the wet soap scrubbing method [16]. A two-gallon pump sprayer was filled with water and approximately 10 mL Dawn^®^ (Procter and Gamble) dish soap. Gear was rinsed with water, sprayed with the mixture, scrubbed with an industrial scrub brush, and given a final rinse until suds were removed. This cleaning practice concluded with hanging the gear in a vented storage container at ambient conditions until the next exposure trial. PPE components remained connected for on-scene wet soap PER. All treatment groups (zipper/laundered, zipper/PER, and hook and dee/laundered) went through 40 exposure trials and cleaning processes. For the off-gassing experiment, a comparison ensemble of PPE that had been through a PER process with only a dry brush after each of the 40 exposure scenarios was included.

### 2.4. Wipe Sampling

Wipe sampling of the turnout gear occurred before and after exposure for burns 1, 10, 20, and 40. Samples collected before the first burn provide a measure of contamination present on new gear. Other “before-fire” samples provide an indication of the effectiveness of the cleaning treatment (i.e., laundering or PER, including 20+ hours of drying) over the study duration (with repeated exposure and cleaning).

The PPE sampling wipes (Allegro^®^ 3001 cleaning pads, 12.7 cm × 20.3 cm) used in this study contained 0.4% benzalkonium chloride solution as a wetting agent. Investigators wearing a clean pair of nitrile gloves used paperboard templates measuring 100 cm^2^ to facilitate collection of wipe samples from PPE surfaces, utilizing two consecutive wipes and wiping in three crisscrossing directions. Field blank samples were also collected to account for any onsite contamination in sample handling. Wipes were analyzed using NIOSH method 5506.

PPE sampling occurred at the front stomach region of the jacket outer shell, jacket liner (also stomach region), and lower front of the outer shell of the pants. Additional wipe samples were collected after smoke exposure from composite fire helmets that were new at the start of the study.

To explore wipe sampling collection efficiency, a commercially available aluminum baking sheet was utilized to provide a nonporous surface that should provide maximal recovery. The baking sheet was placed in the exposure chamber (at the same time as mannequins dressed in PPE) on a tripod at chest-height with a 90-mm polytetrafluoroethylene (PTFE) membrane filter attached to it with tape, creating an area of 6.4 cm × 6.4 cm to capture exposures (e.g., particle deposition) (Appendix A). After the exposure period, the surface of the metal baking sheet next to the filter was wiped in the same format as the PPE (two consecutive wipes, 100 cm^2^) (Appendix A). However, in the first burn, the two consecutive wipes were analyzed separately for PAHs. By doing this, we found that the second wipe was still able to collect a median of 3.42 ug of PAHs (11% of the first and second wipe combined). Hence, we decided to collect and analyze two wipes together for all sampling sites. Wipe sampling was part of a series of measurements and took approximately 10–30 min to complete; therefore, it is possible that more volatile contaminants on the PPE evaporated prior to collection.

In further preparation for this study, 70% isopropyl alcohol wipes (Allegro 1001) were tested against the benzalkonium chloride wipes for collecting PAH contamination from turnout gear. The benzalkonium chloride wipes performed the same or better than the isopropyl alcohol wipes at collecting PAHs (Appendix A). Therefore, only benzalkonium chloride wipes were used to sample PPE surfaces in this study.

### 2.5. Filter Sampling

To evaluate contamination of particulate phase PAHs on surfaces or clothing under turnout gear, 90-mm PTFE filters were affixed to the mannequins. Filters were attached using tape for under the hood (neck region) and inner wrist locations and attached using an alligator clip to the base layer shirt for the chest location on the mannequins. To isolate particulate loading to only the outward facing side of the filter, aluminum backing was applied to the backside of the PTFE filters that were affixed to the base layer shirts. The filter collection area was approximately 6.4 cm x 6.4 cm. Following each trial, PPE was removed from the mannequins and researchers donned clean nitrile gloves to remove the filters and place them into 50-mL opaque Falcon tubes. The samples were stored in a freezer and later analyzed using NIOSH method 5506 [32].

### 2.6. Off-Gas Sampling of PPE

To pilot test the off-gassing potential of PAHs on turnout gear over a 24-h period, three configurations of turnout gear jackets, pants, and gloves at the end of the study (exposed/laundered, exposed/dry brush PER, exposed/wet soap PER) were placed in an air-tight container (Pelican^®^ cases, 0.16 m^3^ volume) for PAH sampling using air samplers (13–8 × 75-mm glass OVS-XAD-7). Air concentrations of PAHs in an empty container were also measured for comparison. The OVS-XAD-7 tubes were analyzed using NIOSH Method 5506 (NIOSH, 2013).

### 2.7. Data Analysis

The descriptive comparisons for PPE surface contamination were carried out using total PAHs, which was the sum of the 15 quantified PAH concentrations (µg/2 wipes) [33]. Non-detectable concentrations below the limits of detection (LOD) were assigned values using the β-substitution method [34] conducted via R version 4.2.0 (R Foundation for Statistical Computing, Vienna, Austria) [35]. Descriptive statistics are presented as median and range for the total PAHs, stratified by PPE ensemble component (exterior jacket, liner jacket, and exterior pant), treatment (laundered, wet soap PER), and timing of sample collection (before fires, after fires). Pie charts display percentages of total mean PAH concentrations on wipes and PFTE filters from exterior of turnout gear. To quantify the wipe collection efficiency, ratios of total PAH concentration on wipes from the aluminum baking sheet to total PAH concentration collected on the filter were calculated, and the corresponding median and range were provided.

For hypothesis testing, a maximum likelihood estimation method via SAS procedure LIFEREG was conducted to determine if the total PAH levels between before-fire and after-fire by sampling site and treatment were significantly different [36]. However, statistical power was insufficient to examine the median differences in before-fire total PAH concentrations between laundered gear and PER-processed gear overall or by PPE component. All statistical tests were two-sided at the 0.05 significance level. Analyses were performed in SAS (version 9.4, SAS Institute, Cary, NC, USA).

## 3. Results

### 3.1. Total PAH Concentrations Wiped from Gear after Repeated Exposure and Cleanings

Regression analysis indicated that there was no effect of repeated exposures (with cleaning between exposures) on PAH contamination levels (all sampling sites combined; before fire 10, 20, and 40; after fire 1, 10, 20, and 40; *p*-value > 0.05, data not shown). We also compared jacket liner results, stratified by jacket closure type (zipper vs. hook and dee) and there were no significant differences (*p*-value > 0.05, data not shown) in PAH concentrations on the liner based on closure type. Additionally, there were no differences in PAH contamination of any of the PPE components between the two jacket closures (zipper and hook and dee). As such, we opted to report results from both jacket closure types in one table, stratified by sampling site (jacket outer shell, jacket liner, and outer shell of the pants), treatment (laundered or wet soap preliminary exposure reduction practice), and collection period (before or after fires) (Table 1).

The differences in PAHs measured on laundered versus wet soap PER gear before fires (across all sampling sites) were significantly different from zero (Wilcoxon signed-rank test, *p*-value = 0.008), suggesting that laundering is more effective at removing PAHs than wet soap PER. We found a significant reduction in PAH levels on the exterior of the jackets (median difference = 17.1 µg/100 cm^2^; *p*-value < 0.001) and pants (median difference = 4.41 µg/100 cm^2^; *p*-value = 0.016) for samples taken post exposure and prior to laundering compared to samples taken after laundering. Similarly, on-scene PER significantly reduced the levels on the exterior pants (median difference = 5.92 µg/100 cm^2^; *p*-value = 0.004) and helmets (median difference = 68.6 µg/100 cm^2^; *p*-value < 0.001). On-scene PER also appeared to reduce the PAH levels on the exterior jacket when sampled before the subsequent fire exposure, but the reduction was not statistically significant (*p*-value = 0.173). Not surprisingly, the jacket outer shell had significantly higher contamination than the jacket liner (difference = 17.0 µg/100 cm^2^; *p*-value < 0.001).

Figure 2a displays the percent of the total mean PAH concentrations for the eight most dominant analytes collected via wipes from the exterior of turnout gear following smoke exposure. Dibenzo(a,h)anthracene was the most prominent PAH detected (31.4%), followed by fluorene (14.9%), and phenanthrene (11.6%); other PAHs, such as benzo(b)fluoranthene, accounted individually for less than 7%, on average, of the total PAH concentrations. Interestingly, naphthalene accounted for less than 1% of the total mean PAH concentrations; this is likely due to vaporization of this compound from the PPE surfaces prior to sample collection.

Figure 2b serves as a comparison to Figure 2a, displaying the percent of the total mean PAH concentrations for the same eight most dominant analytes collected via PTFE filters affixed to the exterior of turnout gear during smoke exposure. Phenanthrene was the most prominent PAH detected (40.2%), followed by naphthalene (19.5%), fluorene (11.6%), and fluoranthene (11.2%). The differences in PAH composition between wipe samples and PTFE filter samples supports the theory that some of the more volatile PAHs were vaporizing prior to wipe collection. Field blanks underwent laboratory analysis and were below the level of detection.

#### 3.1.1. Toxic Equivalency Factors

Toxic equivalency factors (TEFs) were assigned to the eight most dominant analytes collected via wipe and PTFE filter samples using benzo(a)pyrene as the referent (TEF = 1) [5]. After calculating TEQs, we found that dibenzo(a,h)anthracene (TEF = 5) comprised the majority of the total mean TEQs collected on wipe samples (95.1%) and PTFE filter samples (89.6%). Benzo(a)pyrene was the second most dominant PAH, comprising 3.7% and 7.6% of the total TEQs collected from wipes and filters, respectively.

#### 3.1.2. Wipe Collection Efficiency Pilot Experiment

The wipes from the aluminum baking sheet (N = 4) were able to collect a median of 145% (range of 90–180%) of what was collected on the filter. Collection efficiency of wipes from porous turnout gear textiles is likely below what we determined from the aluminum baking sheet experiment. However, this experiment does show that these wipes perform as intended at collecting PAH contamination from surfaces (Appendix A).

#### 3.1.3. Off-Gassing Pilot Experiment

Table 2 summarizes the off-gassing concentrations of PAHs from the three different configurations of turnout gear placed in empty containers for 24-h compared to an empty container (*n* = 1 for each container). Total PAHs are reported, along with the four most dominant contaminants: naphthalene, fluorene, phenanthrene, and pyrene. As expected, exposed gear that went through a dry brush only PER process resulted in the highest total PAH air concentrations (74.5 µg/m^3^), followed by wet soap PER-processed gear (57.8 µg/m^3^) and laundered gear (47.7 µg/m^3^), all of which were well above the levels measured from the empty container (9.6 µg/m^3^). Naphthalene was the most prominent PAH across all conditions.

## 4. Discussion

This project explored the effectiveness of routine cleaning (PER or laundering) of firefighter PPE after repeated exposure to combustion products (i.e., PAHs), as well as the distribution of contamination across PPE sampling sites and the impact of repeated exposure and laundering on contamination. We additionally explored off-gassing of semi-volatile PAHs from used PPE with different cleaning treatments.

Our data show that there are significant differences in PAH surface contamination between gear that was laundered and gear that went through a wet soap PER (*p*-value = 0.008), with laundered components having less PAH contamination overall. Routine laundering lowered median PAH contamination levels (by comparing post-fire values to pre-fire values) on the outer shell of the turnout jackets by >90%. Keir et al. [6] reported that laundering turnout gear reduced PAH surface contamination by an average of 61% and Mayer et al. [25] found that laundering knit hoods removed an average of 76% of total PAHs. However, Banks et al. [23] found that laundering was not effective at removing PAHs from turnout gear ensembles, with evidence of cross-contamination during the laundry cycle. Variability in study results could be due to differences in fuel loads (and hence PAHs), number of exposures, sampling methods, sampling sites, and precise laundering techniques. For example, Mayer et al. [25] and Banks et al. [23] utilized destructive sampling methods, whereas Keir et al. [6] and researchers in the present study used surface sampling methods that are unlikely to capture contamination that is embedded deep into the fabric (but does characterize surface PAHs that may be easily transferred through cross contamination when PPE is contacted).

The PPE jackets utilized for this study had two different closures: (1) zipper and Velcro^®^ and (2) hook and dee. In our study, the type of gear closure did not play a significant role in the PAH-contamination of the jacket liner, and hence, results were not presented by jacket closure. PAH contamination on the jacket liner was significantly lower than the outer shell of the jacket, suggesting that the turnout jackets successfully reduced contamination reaching the liner. Stull [37] emphasized that exposure penetration pathways of PPE occur anywhere that air can move, such as overlaps in articles of clothing and the interface between the hood and collar, and Mayer et al. [25] found that PAH contamination levels in the chest region were 1.5-fold higher among those wearing turnout jackets with hook and dee versus zipper closures (in a companion study using the same mannequins as the present study). Hence, some PAH contamination did penetrate the PPE interfaces even if the jacket liner remained relatively clean, since contamination may not be evenly distributed across a PPE component. Particle physics likely plays an important role in these findings. In addition, vapor phase PAHs such as naphthalene could have condensed to the liner but not been measured because of their tendency to vaporize.

The current study design also allowed us to evaluate the impact of repeated exposure and cleaning on accumulated contamination levels (Appendix A). There was no effect of repeated exposures and cleaning on the contamination levels measured using wipes and PTFE filters (*p*-values = 0.076 and 0.354, respectively), suggesting that routine laundering or PER will help manage exposure risk by at least preventing accumulation of PAHs on turnout gear surfaces with use over time. Our data suggest that laundering is a more effective exposure mitigation strategy than PER alone. However, laundering after every fire could present logistical challenges for some departments, and Horn et al. (2021) found that laundering gear 40 times resulted in some textile properties (i.e., outer shell trap tear strength and seam strength) falling below NFPA 1971 requirements [22].

The composition of PAHs measured from gear differed by sampling method (wipes vs. filters). Dibenz(a,h)anthracene, fluoranthene, and phenanthrene were the dominant species collected with wipes; while phenanthrene, naphthalene, and fluorene were the dominant species collected on filters. We suspect that some of the lower molecular weight compounds (e.g., naphthalene) vaporized prior to the wipe collection. On the basis of relative toxicity (TEQ analysis), dibenz(a,h)anthracene appeared to be the most dominant PAH contaminant measured from turnout gear regardless of the sampling method. Numerous other studies have measured high levels of dibenz(a,h)anthracene relative to other PAH contaminants on firefighter PPE (often representing one of the three most dominant PAH compounds) [25]. It should be noted, however, that firefighters respond to fires that contain a wide range of burning materials. It is likely that actual fires will present a different mixture of PAHs than was produced in this study by repeatedly burning the same fuel.

Many PAHs are considered SVOCs, although PAHs with five rings or more have relatively low vapor pressures (e.g., dibenz(a,h)anthracene vapor pressure = 1 × 10^−10^ mm Hg at 20 °C). VOCs (e.g., benzene, toluene, styrene) on turnout gear will likely off-gas within an hour or less, while SVOCs can be expected to take much longer to vaporize depending on their vapor pressures and environmental factors [19]. As previously mentioned, our wipe sampling method measured surface contamination and not deeply embedded contamination. We found that even after laundering and PER cleaning with wet soap or dry brush, used turnout gear will release some of these embedded contaminants (dominated by naphthalene) into the atmosphere as vapor, potentially exposing fire personnel in settings beyond the fire scene. This potential source of exposure is concerning because the contaminants on, or embedded in, PPE may follow the firefighter into unintended settings, such as the firefighter’s personal vehicle, apparatus cabs, fire station living areas, or home, depending on gear transportation and storage procedures [23]. Our pilot testing results (*n* = 1) show that off-gas concentrations of PAHs were higher for wet soap PER-processed turnout gear than laundered gear, providing some evidence that laundering is the more effective practice for removing PAH contamination, particularly contaminants in the deeper recesses of the textiles. However, laundering does not appear to completely remove PAHs or other higher molecular weight compounds [23,24,25]. Although we found no effect of repeated exposures on turnout gear outer shell contamination, it is possible that multiple uses could result in accumulation of contaminants that are embedded in the PPE fabric. However, our sampling methodology did not allow us to evaluate this. More research is needed.

Previous studies involving non-destructive sampling of firefighter turnout gear have noted the limitation of unknown collection efficiency of surface wipes [16,21]. Our collection efficiency pilot experiment showed that, compared to PTFE filters, a median of 145% of PAHs on a soot-contaminated smooth surface were collected using two consecutive wipes (where the second wipe collected ~11% of the total of both wipes). We used 0.4% benzalkonium chloride wipes for sampling, but 70% isopropyl wipes should perform similarly based on our preliminary experiments (Appendix A). The cause for greater than 100% collection efficiency is unknown but could be due to the wipes picking up additional contamination on the fringes of the 100 cm^2^ templates, effectively resulting in a larger sampling surface area (see Appendix A). It is also possible that some contamination on the filters was lost when removing the tape (see Appendix A). The collection efficiency of wipes will likely vary based on the type of surface being sampled. For example, wipe sampling will be more efficient on smooth composite helmets than porous turnout jackets or pants. However, PAHs not collected by wipes are likely embedded into the fibers and would present less of a cross-contamination concern. Hence, wipe sampling is still a valid technique for assessing cross-contamination exposure risk.

The NFPA 1851 standard requires personnel to evaluate PPE following use to assess the level of cleaning necessary [26]. Historically, in regard to PER and laundering, helmets have been an overlooked part of the PPE ensemble. However, our data revealed a median post-fire PAH contamination level of 76.97 µg/100 cm^2^ on helmets, demonstrating the clear need for cleaning and precautions when handling helmets post-exposure. The helmet exposure in our study may have been higher than during actual suppression activities because the mannequins were in a standing position. Nonetheless, these findings reinforce the need to routinely consider cleaning helmets.

The timing of the collection is an important factor that may impact what is measured via wipe sampling. Wipe sampling of turnout gear found lower naphthalene levels than anticipated based on previous research [38]. While naphthalene is a common PAH in the fire environment, it is typically present in the gas phase as opposed to particulate phase. We speculate that naphthalene (the most volatile PAH) is underrepresented because it vaporized prior to sample collection due to the time it took to collect the samples (~10–30 min). That naphthalene constituted 0.45% of total PAHs collected via wipes (Figure 2a) and 19.5% collected by filters (Figure 2b), which were processed more quickly, further supports this postulation. Kirk et al. [13] found naphthalene to be absent in their sampling of PPE swatches with a similar rationale of the cause.

A limitation of this study was the collection of data from stationary mannequins and vertical distribution of contaminants in the exposure chamber, as opposed to moving human subjects performing firefighting tasks at different locations in a structure. When collecting data in a research environment, it is important to note that testing is occurring under best case scenarios for PPE compliance. For example, the mannequins’ PPE was consistently checked for proper fit with all components fastened appropriately. It is possible that in a real-life scenario, firefighters may not check their PPE for proper closures to the level of detail possible in a simulated situation prior to entering a contaminated area. It is conceivable that breakthrough levels of contamination (captured on the jacket liner) would be higher for a real firefighter who is responding to, and actively moving through, a fire scene. On the other hand, crouching or crawling inside a structure below the smoke layer, as is common during fire responses, would likely result in a lower burden of contamination to the gear (especially the helmet), with the possible exception of pants, boots, and gloves, on which transfer of surface contamination could occur. Finally, this study was also limited to a single fuel load, a commercially available couch, and it is expected that firefighters could encounter a wide variety of burning materials during actual responses. Despite these limitations, using the FES, we were able to expose firefighter gear to repeatable burns that included foams and upholstery that are an important source of PAHs and thus investigate the effect of cleaning techniques on PAH contamination.

## 5. Conclusions

Firefighters’ PPE will become contaminated with a variety of PAHs after use in structure fires. For standing mannequins, the median post-fire contamination levels were highest for the helmets, followed by the outer shell of the turnout jackets and then the pants; the inner jacket liner remained relatively free of contamination. On a toxicity equivalency basis (using benzo(a)pyrene as the referent), dibenz(a,h)anthracene was the dominant species contaminating the PPE after these fires. Both laundering and wet soap PER methods (post-fire) are effective in reducing surface contamination and appear to prevent accumulation of contamination after repeated exposures. However, SVOCs deep within the fibers of bulky PPE may not be effectively removed with PER, or even with machine laundering practices, and could expose fire personnel via off-gassing. Further research is needed to identify the most effective laundering methods for firefighter turnout gear that considers the broad spectrum of common contaminants, including synthetic compounds such as PFAS, halogenated flame retardants, and phthalates. Such research should involve PPE used in emergency fire events to most accurately determine efficacy of decontamination practices.

## Figures and Tables

**Figure 1 ijerph-20-02108-f001:**
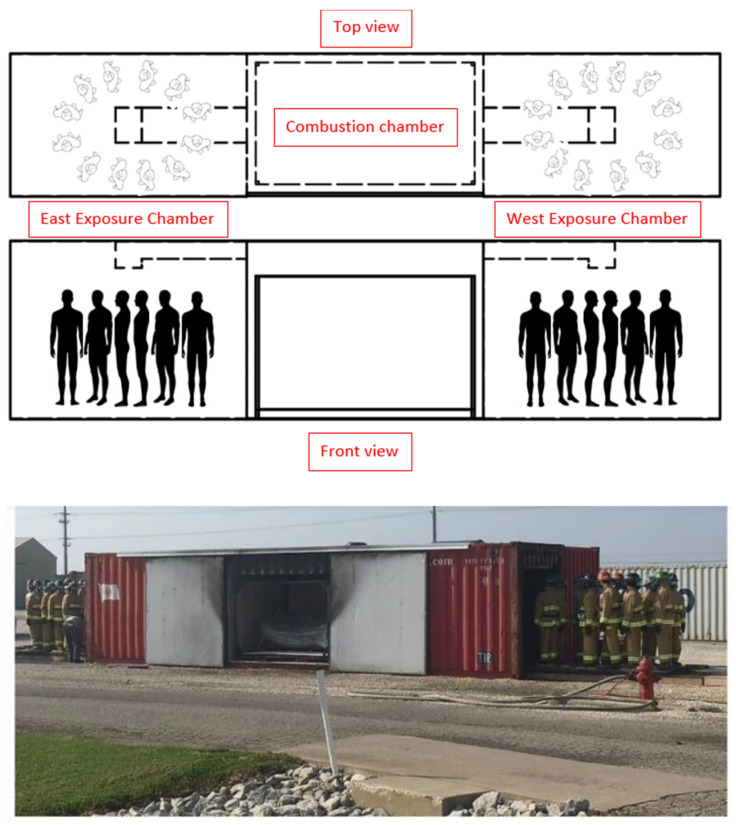
Fire Exposure Simulator.

**Figure 2 ijerph-20-02108-f002:**
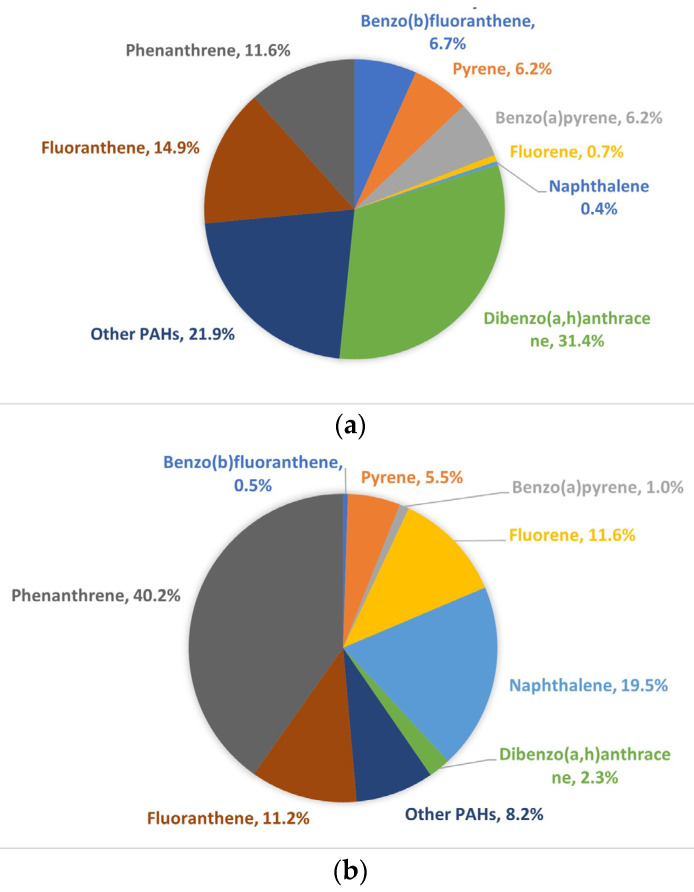
(**a**) Percent of total mean PAH concentrations on wipe samples collected from outer shell of gear and fires (N of jackets = 15 and N of pants = 5).; (**b**) Percent of total mean PAH concentrations on PTFE samples attached to exterior gear and collected after fires (N of jackets and N or pants = 9).

**Table 1 ijerph-20-02108-t001:** Median total PAH concentration (µg/100 cm^2^) on two consecutive wipes of firefighter gear after repeated exposures and cleanings *.

Sampling Site	Treatment	Collection Period Relative to Smoke Exposure	N	N of Non-Detects	Median	Range	*p*-Value ^†^
Jacket Outer Shell (N = 21)	Laundered	Before	3	0	1.55	1.51–1.55	Reference
		After	6	0	18.65	10.73–29.04	<0.001
	Wet Soap PER	Before	3	0	4.63	4.45–6.69	Reference
		After	9	0	8.23	2.02–29.93	0.173
Jacket Liner (N = 19)	Laundered	Before	4	4	<LOD	<LOD	Reference
		After	8	5	<LOD	<LOD–1.12	Not Applicable
	Wet Soap PER	Before	3	0	1.20	0.53–2.32	Reference
		After	4	1	1.81	<LOD–1.86	Not Applicable
Pant Outer Shell (N = 11)	Laundered	Before	3	1	0.27	<LOD–0.81	Reference
		After	3	0	4.68	1.87–5.41	0.016
	Wet Soap PER	Before	3	1	1.26	<LOD–1.43	Reference
		After	2	0	7.18	4.10–10.25	0.004
Helmet (N = 13)	Wet Soap PER	Before	6	0	8.39	1.41–26.10	Reference
		After	7	0	76.97	51.42–153.3	<0.001

* Sampling of gear new before the first fire measured <LOD and were excluded from these data. ^†^ Non-detectable concentrations below the limits of detection (LOD) were assigned values using the β-substitution method that adjusts each non-detectable value based on the uncensored data distribution (Ganser and Hewett, 2010) [34]. Maximum likelihood estimation method (Helsel, 2006) [36] via SAS procedure LIFEREG was utilized to determine if the total PAH concentrations between before vs. after by sample site and treatment were significantly different. *p*-value was specified as “Not Applicable” when the analysis modeling was not convergent.

**Table 2 ijerph-20-02108-t002:** Off-gas PAH concentrations (µg/m^3^) for the four most dominant analytes (Fluorene, Naphthalene, Phenanthrene, and Pyrene) measured from containers (N = 1 type of gear or no gear per container) over 24 h.

Contents Inside Containers	Total PAHs	Naphthalene µg/m^3^	Fluorene µg/m^3^	Phenanthrene µg/m^3^	Pyrene µg/m^3^
Exposed/ Dry Brush PER	74.47	65.04	3.56	4.31	0.35
Exposed/ Wet Soap PER	57.80	48.56	2.66	4.32	0.26
Exposed/Laundered	47.69	39.49	2.93	4.10	0.16
Empty	9.57	8.25	0.17	0.15	0.00

## Data Availability

All available data have been reported in this manuscript.

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
