# Peer review of "Use of Preliminary Exposure Reduction Practices or Laundering to Mitigate Polycyclic Aromatic Hydrocarbon Contamination on Firefighter Personal Protective Equipment Ensembles"

_ijerph, 2023, doi:10.3390/ijerph20032108_

Round 1

Reviewer 1 Report

The article presents strategies to decontaminate personal protective equipment after firefighting operations. 

It is noteworthy that the authors have provided a significant work in terms of experiments, but they could be more clear regarding the conduction of the experiments, why did they chose some procedures, why the p-values were so different?

It is not explicit whether a benchmarking exist for PPE decontamination.

Also, it is not clear some technical procedures and normative adopted by the researchers. 

I have provided some observations  and suggestions in the original file (attached).

Some questions that remain:

- Why did the authors  selected the described methods to expose the PPE to contaminants? 

- Are such methods and scenaries close to the reality?

- Could the results be different with other sample sizes?

- What are the main health effects caused by the contamined PPEs and could they be prevented by alternative or accessory firefighting resources as Artificial Intelligence, drones etc? 

- What are the costs of the alternative contaminant removal choices? 

I understand that the paper is valuable regardless the limitation described in the Discussion section, but some parameters  and choices could be more clearly expressed.

With respect to the title, I suggest to avoid acronyms.

The end of the Introduction must provide information about the structure of the paper.

The procedures could be followed by picures or schematic explanation in order to improve the clarity of the text.

Author Response

Reviewer 1 Comments:
Comment: The article presents strategies to decontaminate personal protective equipment after firefighting operations.
Response: The authors thank you for your thoughtful review.

Comment: It is noteworthy that the authors have provided a significant work in terms of experiments, but they could be more clear regarding the conduction of the experiments, why did they chose some procedures, why the p-values were so different?
Response: The methods for the exposure portion of the study were chosen based off previous experience researching this topic. The fuel load and timing for ignition and suppression have been standardized from work in the fire exposure simulator. The timing has been based on what could be duplicated with human subjects to mimic realistic fire response activity timing. The wipe and off-gassing experiments were pilot projects that were based on feedback from previously published research on the topic. We acknowledge that a lot of experiments were conducted, but we believe the manuscript explains our rationale and how the experiments are connected, particularly in the discussion section.

Comment: It is not explicit whether a benchmarking exist for PPE decontamination.
Response: Although there are national standards (NFPA 1851) which state best practices and verifications for PPE cleaning (as noted in the manuscript), there is no universally required benchmark. A differentiation that needs to be remembered is the difference between preliminary exposure reduction (on-scene decontamination) versus laundering or professional cleaning.

Comment: Also, it is not clear some technical procedures and normative adopted by the researchers.
Response: The methods used have been employed previously (with citations) and additional supplementary data is included to provide even more details regarding collection efficiency. Feedback to individual comments will be addressed in the manuscript review copy.

Comment: I have provided some observations and suggestions in the original file (attached).
Response: Thank you, we have responded in track changes in the manuscript review copy. Comment: Why did the authors selected the described methods to expose the PPE to contaminants? Response: The methods to expose gear were selected based on many years of exposure research using similar methods; this allows for comparisons to be made with previously published work. The use of the particular fuel load is used because the sofa is the most commonly purchased sofa in the United States so we believe it to be a realistic fuel scenario for exposure testing.

Comment: Are such methods and scenaries close to the reality?

Response: We believe these are realistic scenarios. The fuel load (sofa) is the most commonly purchased sofa in the United States. The preliminary exposure reduction (aka decontamination) methods are commonly deployed methods in the fire service. The timing of scenarios is based off of realistic timing for firefighting activities. That said, limitations with respect to having mannequins remain standing in the exposure chamber are highlighted at the end of the discussion.

Comment: Could the results be different with other sample sizes?

Response: The off-gassing experiment (n=1) and wipe experiment were pilot experiments (and are now noted as such in the manuscript) and could be expanded to a larger sample size to determine if results would be affected. However, expanding the sample size will need to happen in future studies.

Comment: What are the main health effects caused by the contamined PPEs and could they be prevented by alternative or accessory firefighting resources as Artificial Intelligence, drones etc?

Response: Previous research has shown that harmful contaminants may be embedded in the fibers of exposed PPE, and/or breakthrough the PPE to firefighter skin. The International Agency for Research on Cancer has identified dozens of chemicals found on the fireground to be known or probable carcinogens, thus increasing the cause for long-term health concerns such as cancer. Yes, if firefighting could be modified for Artificial Intelligence or similar, exposures could be lessened. Drones are currently utilized in some areas of the fire service, but the technology is not yet widely available. Discussion of AI or drones is beyond the scope of the current paper.

Comment: What are the costs of the alternative contaminant removal choices?

Response: The average cost of professional cleaning for PPE is approximately $200 per set of gear. This approach however, does not change the initial exposures and need for preliminary exposure reduction (decontamination) because the exposed PPE will still be widely handled prior to sending out for cleaning. The methods for cleaning discussed in this paper are for on-scene/immediate reduction of exposures. Discussion of costs is beyond the scope of the current paper.

Comment: I understand that the paper is valuable regardless the limitation described in the Discussion section, but some parameters and choices could be more clearly expressed. Response: Thank you for the feedback, we have reviewed the manuscript for clarity and made edits as appropriate. Comment: With respect to the title, I suggest to avoid acronyms.

Response: Thank you, we have updated the title to reflect this suggestion. Comment: The end of the Introduction must provide information about the structure of the paper. Response: We disagree because the paper is organized in a standardized way (methods, data analysis, and discussion). However, if the editors would like us to do this, we would be happy to craft a sentence.

Comment: The procedures could be followed by picures or schematic explanation in order to improve the clarity of the text.
Response: Thank you for the suggestion. We have added an additional supplemental image to address this comment.

Reviewer 2 Report

General comments:

This paper (ijerph-2132611) reported an experimental study on the relationship between contaminated firefighter PPE and potential exposure to PAHs as well as the efficacy of cleaning practices. In general, the paper was well organized and the results of the study are meaningful. I think that this paper is suitable to publish in International Journal of Environmental Research and Public Health after minor revision.

Specific Comments:

Line 55: There should be a full name of PER for the first presence in the text.

Line 60: “hazard” may be “hazards”.

Line 126: “(FSRI” should be “(FSRI)”.

Lines 192 and 208: “cm2” should be “cm2”.

Figure 2. The text shown in the Figure 2 should be enlarged, the current words cannot be seen clearly.

Lines 536, 552, and 566. The journal names are not abbreviated. Please check and correct the format of the references.
